# Assisting Maritime Search and Rescue (SAR) Personnel with AI-Based Speech Recognition and Smart Direction Finding

**Aylin Gözalan** [1], **Ole John** [2], **Thomas Lübcke** [1,*], **Andreas Maier** [3], **Maximilian Reimann** [2], **Jan-Gerrit Richter** [4] and **Ivan Zverev** [3]

1    German Maritime Search and Rescue Service—DGzRS, Research & Development, 28199 Bremen, Germany; goezalan@seenotretter.de
2    Fraunhofer Center for Maritime Logistics and Services CML, 21073 Hamburg, Germany; ole.john@cml.fraunhofer.de (O.J.); maximilian.reimann@cml.fraunhofer.de (M.R.)
3    RHOTHETA Elektronik GmbH, 82418 Murnau, Germany; andreas.maier@rhotheta.de (A.M.); ivan.zverev@rhotheta.de (I.Z.)
4    Fraunhofer Institute for Intelligent Analysis and Information Systems IAIS, 53757 Sankt Augustin, Germany; jan-gerrit.richter@iais.fraunhofer.de
*    Correspondence: luebcke@seenotretter.de; Tel.: +49-42-15-370-7451

**Abstract:** Communication for processing relevant information plays a paramount role in developing a comprehensive understanding of Search and Rescue (SAR) situations and conducting operations in a successful and reliable manner. Nevertheless, communication systems have not changed considerably in the context of simplifying very high frequency (VHF) maritime communication and enhancing the value of SAR practices. The Automated Transcription of Maritime VHF Radio Communication for SAR Mission Coordination (ARTUS) project approaches this problem with the development of an assistance system which employs AI-based speech recognition and smart direction finding. First, ideas and specified needs of end users for designing the user interface are presented in this paper. Further, preliminary accomplishments of domain specific language training for maritime speech recognition, and the direction-finding algorithms for localizing senders are sketched out. While the preliminary results build a solid ground, additional field experiments will be conducted in order to enhance the accuracy and reliability of speech recognition and direction finding. The identified end user requirements across different personnel groups show commonalities, but call for a differentiated approach in order to meet the challenges and peculiar needs of the various working contexts.

**Keywords:** maritime search and rescue; VHF maritime radio; coastal radio station; speech recognition system; direction finding; AIS; maritime simulation; training; GMDSS

## 1. Introduction

Maritime very high frequency (VHF) radio is a robust and reliable communication system that has been established in international shipping for decades and is mandatory on board Safety of Life at Sea (SOLAS) ships in all sea areas [1,2]. Looking at the dynamic change in information systems on board ocean-going vessels, comparatively little has changed in the basic functionality of VHF radios to date—one could almost say that if there was something better, it would have been established long ago. But just as long as VHF radio has been in use, users have been confronted with significant disadvantages of this technology, such as the limited acoustic intelligibility of voice messages or loss of information content [3]. Practice has come to terms with these limitations and coping technologies such as last-call repeat have compensated for system limitations in certain areas.

The limitations of VHF radio are sometimes particularly visible in certain working areas and situations, including in coastal radio stations where human radio operators are listening on several relays simultaneously and/or in maritime search and rescue (SAR) operations which are characterized by a considerable communicative workload. In these settings, maintaining an overview, selecting, and documenting all essential information is a challenge that is currently technically unsolved [4]. In particular, on board small rescue units (rigid hull inflatable boats, fast rescue boats), the noise exposure from engines and airflow, whole-body vibrations, as well as the lack of suitable writing surfaces, make the perception and recording of essential radio communication difficult. Supporting the work of SAR personnel with an adequate assistance system was the motivation for the ongoing research projectARTUS, whose system approach and preliminary results are described in this paper.

## 2. Materials and Methods

The research project is dedicated to the development of an assistance system to support maritime radio communication with a user-centered approach. It combines a real-time speech recognizer for maritime VHF radio communication with a novel radio direction-finding system whose localization algorithm uses additional data available on board to determine the location of the transmitter of a radio message. With the ARTUS system, it will be possible to automatically annotate received radio messages and simultaneously localize the transmitter. This will allow rescue units to find people in distress faster than before, even if contact with them should break off.

In order to develop the system as close as possible to practical application, end users were recruited to define in the first step several potential applications from the field of maritime search and rescue services, in which the target technology aims to provide essential working solutions. Based on these use cases, future user groups for the technology were then selected: radio operators in coastal radio stations, duty officers, and SAR mission coordinators of a Maritime Rescue Coordination Center (MRCC), crew members of lifeboats and rescue cruisers, as well as instructors involved in SAR simulator training. In moderated focus groups with selected experts from the respective application contexts, a catalogue of basic requirements was formed for the development of the various demonstrators of the ARTUS system. Section 3.1 presents the results of this first focus group work. Further iterations are planned in this project to ensure the maturity and practical suitability of the developed demonstrators.

Parallel to the creation of the above-mentioned items and a more detailed requirements catalogue, the compilation of training data for the speech recognizer was initiated. Within the framework of the project, a bilingual system was to be designed that is capable of interpreting maritime vocabulary in English and German. Two pre-trained speech recognizers, which were previously used for real-time recognition of media content were adapted to the limited audio quality of VHF radio messages as well as the domain-specific vocabulary and the various accents of non-native speakers. The following data sources were used for this purpose:

- Manual transcription of VHF maritime radio communications received and recorded in the German Search and Rescue Region of Responsibility by Bremen Rescue Radio,
- Recording of Standard Marine Communication Phrases (SMCP) [5] by different speakers, and
- Development of a domain specific language lexicon based on written reports, written SAR procedures, and training documents.

First, results from training of the speech recognizer with the above-mentioned data as well as technical challenges for further training are described in Section 3.2. The speech recognizer is complemented by a new localization algorithm, which in addition to the results of Doppler direction finding, also includes received AIS signals from surrounding shipping vessels in the evaluation. On this basis, probability statements on the potential sender of a radio message are made. In Section 3.3, the preliminary hardware configuration of the entire system is shown in a block diagram. Furthermore, insights are given into the current implementation status of the localization algorithm.

However, the work status presented in Sections 3.2 and 3.3 can only be regarded as intermediate results on the way to a final system with the required technical maturity and suitability for practical use. Therefore, both the speech recognizer and the localization algorithm will be further developed iteratively in isolation. This will be followed by an integration and testing phase in a laboratory environment, which will be done before the reliability and interaction of the two core components are validated by test setups in different traffic areas and scenarios.

## 3. Preliminary Results

### 3.1. End User Requirements

Considering end users' needs plays a paramount role in innovations that solve problems effectively and create valuable work practices in the long run (e.g., in the maritime domain) [6]. The participation of end users in the development process of new information systems can help to specify system requirements, which determine the success and adoption of new applications. The involvement of end users in designing systems is important for determining which information needs exist for coordinating work activities and making decisions. Hence, the involvement of end users plays a paramount role in understanding how technological systems need to be specified and designed to optimize acceptance and practical workflows for potential users [7].

From this perspective, the views and ideas of the end users were integrated at an early stage of the development phase so that the ARTUS system can be technically realized according to the (information) needs of the end users, and the challenges of their respective working contexts. Specifically, based on prior elaborations, four key user groups were identified from the following working areas:

(1)　On board
(2)　Coastal radio station
(3)　Maritime Rescue Coordination Centre (MRCC)
(4)　SAR Simulator Center.

A moderated workshop was conducted with members of these work domains in order to explore and collect end user centered ideas from different work areas (see Figure 1). Creativity workshops are a widely acknowledged participative approach for exploring and exploiting feasible ideas of relevant stakeholders in the innovation process. Further, workshops serve as an interesting research method. As a research practice, they allow researchers to identify relevant aspects of a certain domain, which is necessary to meet the needs and challenges of complex work contexts. As such, workshops provide a promising avenue for enhancing end user centered research results. Importantly, relevant stakeholders can temporarily participate in the design and research process and enrich technical conceptualizations with their own experiences and work needs [8,9].

The ideas and expectations from the workshop groups were collected, documented, and evaluated in a two-stage process. Firstly, by the moderators of the respective end user groups. Subsequently, these group-focused insights were discussed within the whole research team in order to elicit commonalities and differences between the groups. Additionally, the inter-group reflections facilitated the development of shared understandings with regard to the desired system features and prioritization. Table 1 depicts the ranking of discussed features across all end user groups as a must-have (1), nice-to-have (2), unimportant (3), and disturbing (4).

Classification of the features helped to control expectations among participants and generated a clear focus for the user interface requirements specification. The key findings depicted in Table 1 can be summed up as:

To begin with, the manual or automatic storage of communication transcripts was prioritized by all work groups. Working conditions that necessitate a storage function for all end user groups encompass the need to have 'hard' memory that serves as a valid information source for understanding the current SAR situation. Here, some end users mentioned that a comprehensive and reliable 'picture'

of the overall situation is developed in several phases. This entails the idea that information in the transcripts can be verified and marked with regard to their correctness or accuracy (e.g., using a traffic light system). In this context, automated audio recording of radio messages with a repeat or replay function was also ranked as a must-have in order to verify transcripts or obtain further information or data that was not recognized.

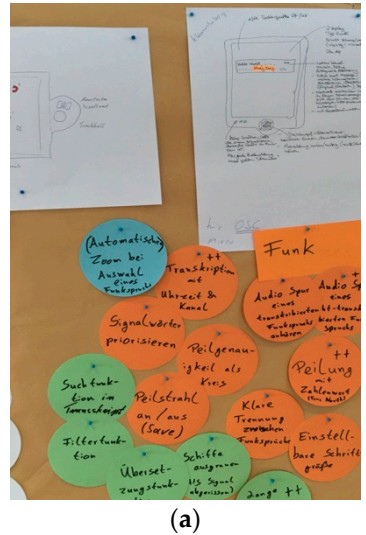
(**a**)

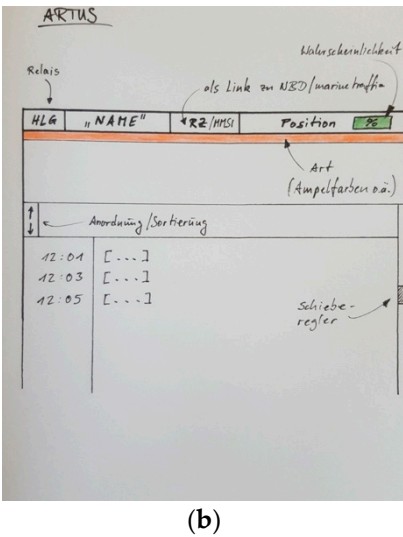
(**b**)

**Figure 1.** Visualized ideas of end users in the onboard group (**a**) and the coastal radio station (**b**).

**Table 1.** Ratings of Possible Features by End User Groups.

| Function | On-Board | Coastal Radio Station | MRCC | SAR Simulator |
|---|---|---|---|---|
| *List view of transcripts* | 1 | 1 | n.a. | 1 |
| *Calls recorder* | 1 | 1 | 1 | 1 |
| *Search/filtering feature* | 2 | 2 | n.a. | 1 |
| *Marking transcripts* | 2 | 1 | 1 | 2 |
| *Translation* | 3 | 3 | 3 | 2 |
| *Display on a sea chart* | 1 | 1 | n.a. | 1 |
| *Direction finding beam* | 2 * | 3 | 3 | 3 |
| *Explicit display of detection possibility* | 2 | 3 | 3 | n.a. |
| *Additional devices* | 1 | 4 | 4 | n.a. |

1 = must-have; 2 = nice-to-have; 3 = unimportant; 4 = disturbing; n.a. = not available/applicable (* *With the option to switch it off. Otherwise, this feature is considered to be disturbing*).

All end user groups elicited the importance of clearly separating incoming radio messages from one another (e.g., by time stamp or the sender of a message). Here, another common denominator between the groups is that small sections of the transcript appear in a chronologically-ordered list view, which can then be fully displayed with additional information. Specific ideas with regard to the list view encompassed the possibility of sorting the list arrangement according to time (arrow) or certain categories (e.g., numbers).

Generally, all groups identified the option to highlight listed transcripts as potentially beneficial although differences can be discerned with regard to its prioritization. For instance, this possible feature to distinguish relevant communication transcripts in the list from unimportant ones is a nice-to-have for board crew members, but a must-have for team members from coastal radio station (e.g., using stars like in 'WhatsApp' chats). Further, in almost all end user groups the possibility to highlight transcripts with regard to their urgency classification (Securité, Pan-Pan, and Mayday) represented a must-have or nice-to-have (e.g., a traffic light system).

Members of the coastal radio station expressed no preferences regarding text font (size) of the transcript apart that it should not 'stick out' from the regular text fonts of the prevalent software. In comparison, members from the board crew and the simulator group expressed the need to be able to adjust the text font size. All end users other than the simulator group seemed to prefer system layouts that 'go along' with their current applications so that relevant information can be transferred and displayed in the current systems for coordination and documentation purposes.

The usability of a potential filter and search function was also stressed across end user groups in order to find and highlight relevant information in the communication transcripts (e.g., call sign, position, persons on board). Here, team members of MRCC and the coastal radio station articulated the need to copy and paste relevant information and data from the ARTUS System to the prevalent software which is used for the coordination and documentation of SAR operations.

Generally, end users preferred that localized ships or identified search areas are displayed on the existing electronic chart display and information systems that are already in use or will be implemented after next year. Mostly, end users considered AIS based data (e.g., call sign, position) and ship name of the localized sender as important. Differences exist about how these are delivered or presented. While some preferred a 'pop up' window for more detailed information, others favored that the relevant information is already given in the list view when listed transcripts are selected. Another interesting idea, which emerged in this context, was to link identified call signs with selected AIS based databases in order collect additional relevant data about the localized senders when needed (e.g., vessel type, length × breadth).

Both the simulator working group and board crew members developed the idea to present localized senders of current communication with a 'blinking' icon. Another interesting finding is that some end user groups (coastal radio station group, simulation, or MRCC) preferred to have no direction-finding beam on the display. Board crew members preferred one, which can be individually switched off when communication density rises. Other differences can be identified with regard to the display of detection probabilities. For instance, probability of detection can be presented as a circular graph according to members of the board crew or as a bar chart. In comparison, other members of the coastal radio station and the MRCC considered detection probabilities as meaningless unless the information is conveyed as an estimated search area or a concrete position on the sea map.

Regarding ergonomics, board crew members for the coordination of operations favored an additional handheld mobile device with buttons and other important 'on board-friendly' settings like daytime and night modes, waterproofness, etc. (see Figure 1). MRCC members and operators of the coastal radio station preferred no additional hardware, which interferes with their current work devices. Particularly, members from the coastal radio station and the simulator group expressed their need to have resizable and moveable windows that can be adjusted on current screens.

### 3.2. Speech Recognition System for VHF Maritime Radio Communication

With the rise of computational power and the use of digital signal processing starting in the second half of the 20th century, research efforts into automatic speech recognition (ASR) began to show promising results. While early systems could only recognize a few words or digits [10], current systems using powerful hardware are able to handle millions of words [11].

In modern systems, three components accomplish speech recognition. The first component, the acoustic model is used to find the highest probable phonemes from acoustic features extracted from the audio. A phonetic pronunciation dictionary is used as a second component to map sequences of phonemes to words. Lastly, a statistical language model finds the most probable sentences from the found sequence of words. The outputs of all components are combined in a global search for the most likely uttered sentences with the audio file as an input.

The acoustic model uses a Time Delay Neural Network (TDNN) architecture with additional projected Long Short Term Memory (LSTMP) layers proposed in [12] together with the LF-MMI criterion proposed in [13]. This architecture was found to outperform other approaches and represents

a state-of-the-art approach in automatic speech recognition [14]. For generation of the phonetic pronunciation dictionary, the open source toolkit Phonetisaurus [15] is commonly used. The language model is based on an n-gram approach [16] with 5-g as the largest word chain. The used ASR system uses the open source tool Kaldi [17], which was originally developed for both German and English broadcast media. At the time of writing this manuscript, English is the main focus of the experiments, with similar adaptations to the system planned for German. The acoustic model of the English system is trained with a combination of datasets from the corpora Librispeech [18], Commonvoice, Fisher, and Switchboard. Overall, the data comprised over 3400 h of annotated speech. Additional to this original data, two augmented datasets were incorporated to account for background noise between 5 and 10 dB and some room reverberation [19]. The system has been successfully adapted to other challenging conditions such as German dialects [20] or oral history interviews [19].

The language model is trained on crawled web data from broadcast media sites and comprises approximately 250,000 unique words. The challenge of performing ASR on VHF Maritime Radio Communication can be split in three different components, each concerning one of the components of the ASR system that needs to be adapted for good ASR quality. For the acoustic model it is necessary to mitigate severe background noise, clipping on the microphone, and a band limitation from the radio. The dictionary, which maps phoneme sequences to words, also needs to be adapted to cope with non-native speakers and varying pronunciations. Lastly, in the language model, the special vocabulary and sentences from the Standard Marine Communication Phrases as well as common ship and location names need to be incorporated.

To build a data basis, approximately 25 h of real radio communication at the Bremen Rescue Radio was manually transcribed. Additionally, approximately 2 h of manually transcribed audio from a simulator are available. This data is used in two ways. Two test sets, each comprising approximately one hour of audio, are randomly taken from the data to objectively evaluate the performance of the ASR system. This corresponds to a 96:4% ratio between training and test data for the real dataset. No data from the simulated dataset is taken into account during the training of the components. As the data from the simulator does not suffer from any of the real-world acoustic drawbacks, the performance of the language model can be evaluated separate from the acoustic model. The test set from the real radio communication is used as a performance measure of the full system.

The remaining real radio communication data can be used to adapt the existing acoustic model to the new conditions. This approach, introduced in [21] and modified in [19], uses large amounts of data from a different domain, and adapts the resulting model with less data from the target domain by fine-tuning the weights of the previously trained neural network acoustic model to achieve better recognition in the target domain.

Figure 2 shows the case insensitive word error rate from both test sets over several development steps.

At the beginning and as a baseline, the data can be evaluated with the language model trained on extracted sentences found in the SMCP only. Approximately 2000 sentences would then be taken comprised of 4000 words.

This results in an expected high error rate, as people generally do not follow the sentences precisely. Additionally, error for the real data is expected to be higher, as errors stem from both the language model and the acoustic model.

The effect of the adaption of the acoustic model can be seen in the next step, as the acoustic model is adapted using the real data. The adaptation has the desired effect of profoundly lowering the word error rate of the real data, as expected.

Lastly, SMCP texts, common ship names, and ship type abbreviation are added to the language model. The ships names are tokenized in a way that the type abbreviation and multi-word names are recognized as one distinct unit, for example W-M-A-Admiral-Jellicoe. This effort reduces the word error rate to approximately 50%.

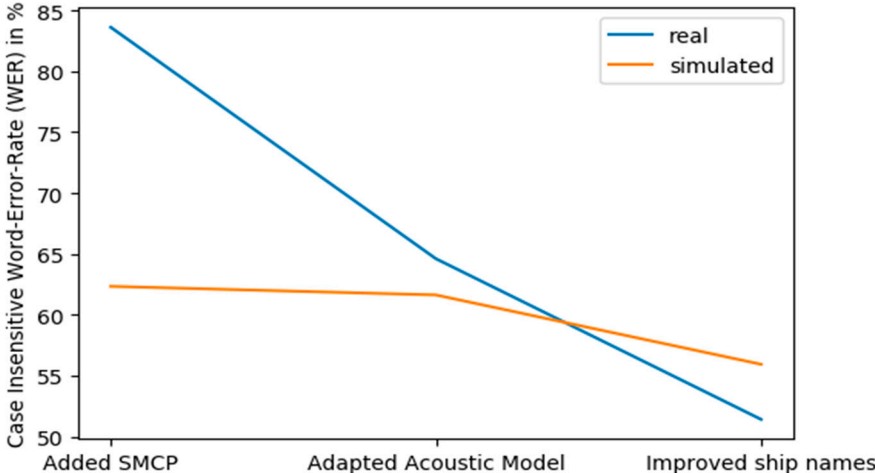

**Figure 2.** Development of the word-error-rate (WER) through the different development steps. Initial values are without adaption, "Adapted Acoustic Model" represents the state acoustic model adaptation, "Improved Ship names" after the addition of typical ship names and ship type abbreviations.

In the future, several further steps will be made to further lower the word error rate. Firstly, a good mix between SMCP sentences and words and non-marine speech needs to be found in the language model, as the used speech is not completely covered by the SMCP. Secondly, a new acoustic model needs to be trained with the acoustic limitations in mind. To this end, the data augmentation needs to be modified to include severe noise and clipping, as well as a narrow band limitation. Lastly, more error reduction can be achieved with the consideration of English accents in the pronunciations. Together, these three measures should be able to again drastically decrease the speech recognition word error rate. The goal of the system is to reduce the Word-Error-Rate to below 20% to ensure acceptable transcription accuracy.

### 3.3. Smart Direction Finding Algorythms

The primary goal of the ARTUS direction finding algorithm is to detect the sender position of a maritime radio message (localization). In addition, the algorithm should also clarify the identity of a potential sender and therefore determine which vessel has transmitted the radio message under consideration (identification).

Different technologies can be used to obtain the location of vessels, with the current most widespread systems in the maritime sector being briefly presented below:

- Radar
- AIS
- Radio direction finding.

The term radar stands for radio detection and ranging. This is a device that actively emits electromagnetic waves in a radius of 360° to localize an object. These waves are reflected from electrically conductive surfaces and can be detected by the receiver. Evaluating the echo provides information regarding distance, direction, and speed of the recognized object [22].

AIS is an abbreviation for *A*utomatic *I*dentification *S*ystem, an international standard for the exchange of navigation and ship data to avoid collisions. With the help of AIS transponders, data between ships is continuously exchanged using the so-called time slot method on two defined frequencies in the VHF marine mobile band. Part of the AIS messages is the dynamically transmitted ship's position, which can be determined at regular intervals using GPS technology [23].

In contrast, in radio direction finding the spatial angle of incidence of a radio wave transmitted by a radio telephone system is measured at a direction-finding antenna. This process of passive measurement makes it possible to determine the direction from which a radio signal originates [24].

In this section, the abbreviation DF is used for both direction finder and direction finding; with the correct meaning from the context under consideration.

The RT-500-M is a Doppler multi-band radio DF which operates on all frequencies between 118 and 470 MHz. The antenna unit of the system is constructed out of two hollow cylinder elements, which represent a thick broad band dipole antenna. When the electromagnetic wave arrives at the antenna construction, it produces a corresponding RF current flow on the antenna surface, which is measured via four symmetrical contacts (North, East, South, and West). Due to the antenna geometry (base diameter), the received signal on every contact has a different phase. This phase difference is the main reason for the emergence of a Doppler frequency modulation at the receiver input when four contacts are switched by the antenna control unit with a frequency of 3 kHz. In other words, the DF system emulates one dipole antenna moving in a circle of the electromagnetic field and modulates the input signal with a FM of 3 kHz. The bearing angle is then calculated by measuring the phase between the FM-demodulated signal and the reference signal, which is used to "rotate" the antenna.

Bearing accuracy of a DF system is not limited by the Doppler principle. It mainly depends on the symmetry of the antenna construction and the internal signal processing. The system accuracy of a RT-500-M direction finder of 5° RMS is guaranteed to be reliable at all bands. However, the total accuracy of an installed DF can be affected by occurring reflections. This is mainly dependent on the system installation environment and especially on the existence of conducting materials in the immediate vicinity of the direction finder antenna.

Based on the previous technology illustration, DF is very suitable for achieving the ARTUS goals. Due to the propagation of radio waves at the speed of light as well as negligibly small distances between sender and receiver of the radio wave, the point in time at which a transmitted radio message is detected by a DF corresponds to the use of the radio device. In an ideal case—i.e., without the occurrence of effects such as reflection of radio waves—the origin of the radio message is thus located directly on the bearing line detected by the DF. This means that a DF can be used to determine both the start of a marine radio call and the bearing line on which the communicating ship is located. With the help of the other technologies presented, these conditions cannot be met at the same time.

When receiving a radio wave, the DF analyzes the electromagnetic wave field that surrounds the DF antenna, which usually consists of several dipole elements. In this way, the relative angle of incidence to the transmitting radio device is determined. In conjunction with other on-board data and intelligent data processing, conclusions can be drawn about the position of a potentially communicating vessel, consisting of its current latitude and longitude. Various hardware components have to be implemented in order to use the localization algorithm. In addition to a DF, a GPS receiver must be used at the antenna installation site. This enables the measured bearing angle to be related to the exact position of the DF antenna and also ensures the time allocation of transmitted radio messages. A north-related bearing can be determined by adding a compass to the system. An AIS receiver provides ship-related information that is required to identify a communicating ship. These include: time stamp of AIS message, ship position, speed over ground, course over ground, rate of turn, call sign, MMSI, vessel type, etc.

Figure 3 is an overview of the hardware components for the on-board demonstrator, which must be installed in order to achieve the project objectives.

The ARTUS algorithm performs its calculations based on the parameters obtained from these hardware components. The basic idea of the algorithm is to link the current position of a vessel under consideration with the direction from which the signal of the radio message originates. Essentially, the following processing steps are carried out in order to locate and identify a communicating ship:

(1)　Detection of the electromagnetic wave of a marine radio message and measurement of the corresponding bearing angle *BRG*.
(2)　Prediction of the current position of a surrounding ship *i*.
(3)　Calculation of the spatial bearing angle $\varphi i$ between own position and the predicted position of the observed ship *i*.

(4)    Estimation of the bearing probability $P_{BRG}$ *(ship$_i$)* based on the determined parameters.

(5)    Repeat steps (2) to (4) for all ships *i* that are within the AIS range.

In step (1), the calculation routine of the DF algorithm is called after a radio signal has been detected by the DF. The measured bearing angle *BRG* serves as a reference for the calculated bearing angle *φi* in the further course of the processing.

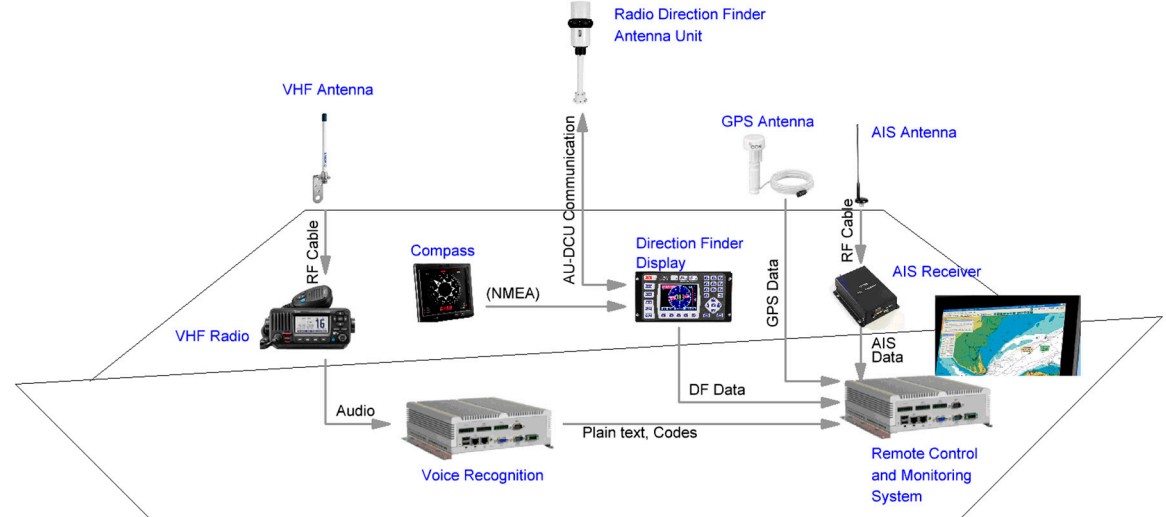

**Figure 3.** Block diagram for the Automated Transcription of Maritime VHF Radio Communication for SAR Mission Coordination (ARTUS) on-board demonstrator.

Since AIS transmission intervals vary depending on the speed and course of a vessel, there is usually a time delay $\Delta t$ between the arrival of the last received AIS signal and the last detected radio signal. This means that the ship continued its voyage in the observed period $\Delta t$ and thus also changed its position. In order to be able to draw meaningful conclusions about the identity of a ship from the measured bearing line, the position of ship *i* at the time the radio signal was received must be used. For this reason, the current position of vessel *i* is predicted in step (2) based on the elapsed time $\Delta t$ and the dynamic ship data contained in the received AIS message. It is assumed that the ship executes a uniform movement starting from its last transmitted position.

After carrying out the previous step, two positions are known: The own position determined by the GPS receiver and the predicted position of the relevant ship *i* at the time the radio message was received. The ARTUS algorithm then calculates the bearing angle *φi* between these two positions in step (4), taking into account trigonometric relationships and Vincenty's formulae. When calculating bearing angles, this formula implies the assumption that the shape of the earth is a rotational ellipsoid.

Figure 4 visualizes the bearing angle *φi* calculated by the algorithm on the basis of two different foreign positions, as well as the bearing angle BRG measured with the DF. Both the calculated and the measured bearing are true north. In example (I) the difference between the measured bearing BRG and the calculated bearing *φi* is small, which is evident from the very similar orientation of the two bearing lines. This results in a relatively high probability that the radio signal was sent from the direction of the calculated bearing line. In example (II), the difference between the two bearing angles is very large, which results in a very low probability that the radio signal was transmitted from the direction of the calculated bearing line.

In step (4), the algorithm computes the so-called bearing probability *PBRG* taking into account the systematic measurement error of the DF and using suitable statistical methods to compare the calculated and measured bearing angles. The bearing probability *PBRG* indicates the likelihood with

which the detected radio signal was transmitted by ship *i* under consideration and can be summarized by the expression in Equation (1): Calculation of bearing probability.

$$PBRG_{Ship_i} = \frac{f\left(\varphi_{Ship_i}\right)}{f(BRG_{DF})} = e^{\left(\frac{\varphi_{Ship_i} - BRG_{DF}}{\sigma_{DF}}\right)^2} \tag{1}$$

Finally, in step (5), the calculation process from steps (2) to (4) is repeated for all ships *iϵI* that are within the AIS range of the own position. The higher the value of the probability, the more likely it is that the ship *i* actually sent the radio message and the smaller the difference between the two bearing angles *φi* and *BRG*. In the ideal case, the value of the calculated bearing angle *φi* and measured bearing angle *BRG* are the same, which means that the bearing line lies exactly on the connecting line between one's own position and the position of ship *i*.

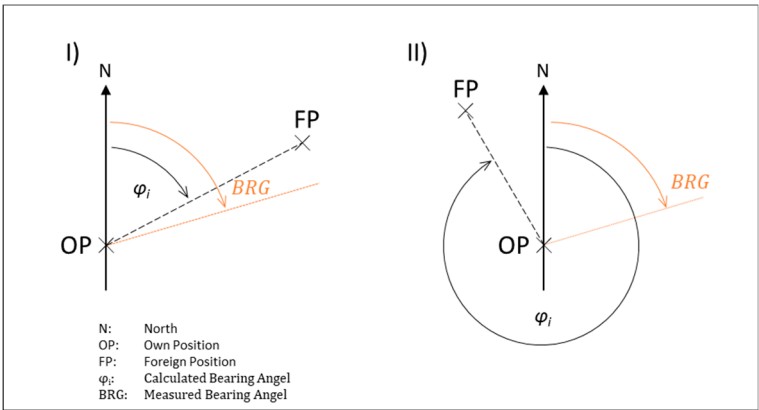

**Figure 4.** Exemplary visualization of the calculated bearing angle *φi* for two different foreign positions: (I) small difference between measured bearing BRG and calculated bearing *φi*; (II) big difference between measured bearing BRG and calculated bearing *φi*.

The pseudocode for determining the bearing probability PBRG can be summarized according to the following formulation:

---

*While* receiving bearing signal *do*:
   *for* all ships i € I within AIS range:
      Determine current position of other ship *i*
      Calculate angle *φi* between own position and ship *i*
      Determine measured bearing angle BRG
      Calculate bearing probability PBRGi

---

The DF algorithm is called again when a new radio signal is detected and then the probabilities are recalculated for all ships within the AIS range.

The dependency of the bearing probability on the measured bearing is shown in Figure 5. A DF is installed on the coastline to determine the bearing angle *BRG* of an incoming radio signal. In addition, there are two vessels in the immediate vicinity of the DF. In situation (a), no vessel is using a radio device so the DF does not receive a radio wave and consequently does not measure a bearing. The DF then receives a radio message in situation (b) and determines a bearing of 0°. This radio message was apparently sent by another ship which cannot be seen on the presented nautical chart. These circumstances are also reflected in the values of the relatively low bearing probability of the two ships displayed, each with 11%. In situation (c) a new radio signal with a bearing of 15° was detected by the DF. Thus, the probabilities of the two depicted ships change and that the ship located east of the DF has a higher bearing probability (40%). Despite the higher bearing probability, this radio signal

was probably sent again from a vessel not shown on the nautical chart. In the last situation, (d) a radio device is used again and a bearing of 40° is measured. This time it is almost certain that the ship on the bearing line is the sender of the radio message. For this reason, it has a bearing probability of 99%.

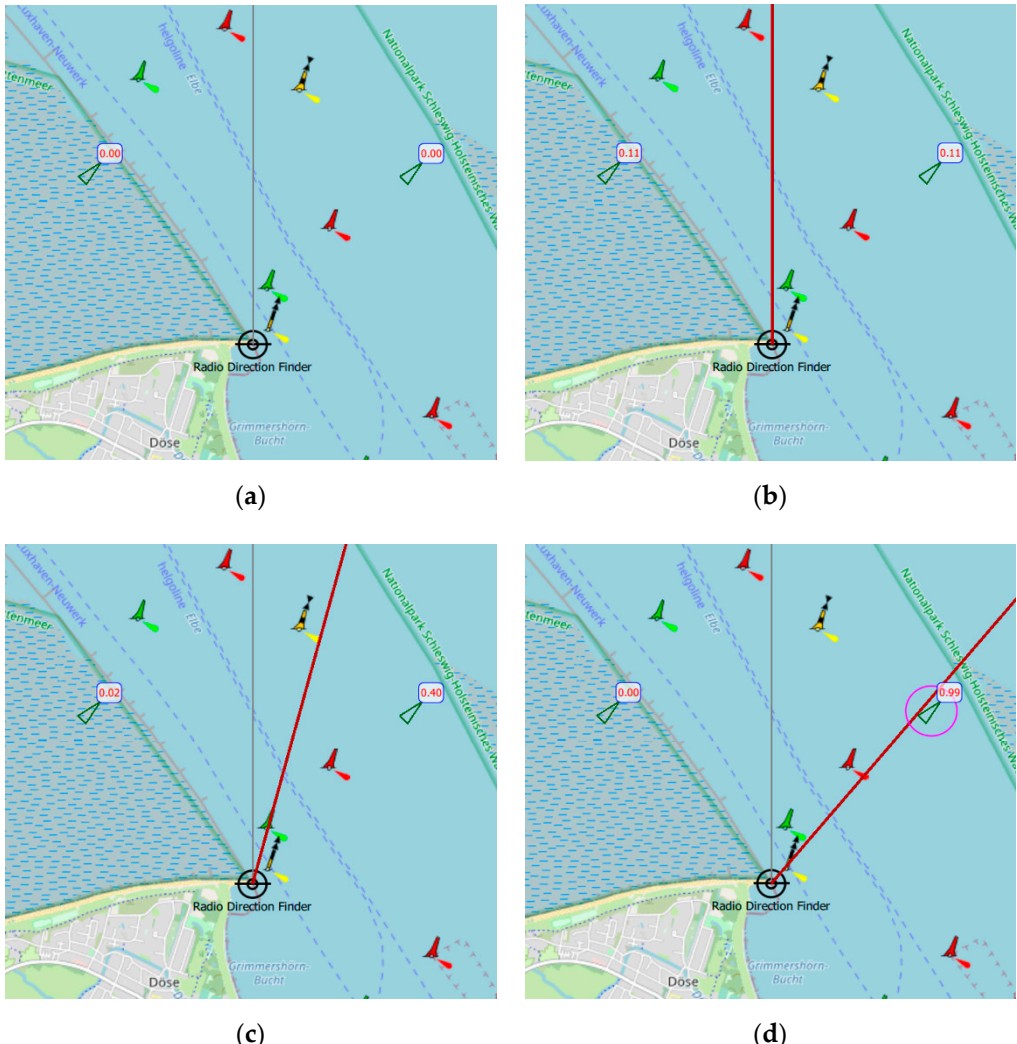

**Figure 5.** Representation of the bearing probability for different measured bearings: (**a**) Radio device not used, no bearing; (**b**) Radio device used, measured bearing of 0°; (**c**) Radio device used, measured bearing of 15°; and (**d**) Radio device used, measured bearing of 40°.

## 4. Discussion

Preliminary results show that the selected target groups have clear ideas about the benefits of ARTUS technology for their respective fields of activity, as well as concrete expectations regarding the required range of functions and ideas for designing a user-friendly human-machine interface. However, the results also show that a one-fits-all solution will not satisfy any of these user groups. Rather, in further development, the aim must be to transfer the specifics of each use context into adequate demonstrators for the respective target groups. A critical point to note is that when experts are involved in the research process, there is always the risk of key informant bias, i.e., reference is made to too few or too homogeneous sources. This was avoided, at least with regard to the homogeneity of the experts involved. Going forward for the project, the demonstrators must be evaluated several times by different end users.

Although the work status achieved so far in the development of a speech recognizer for maritime radio communication established a solid ground, further efforts are needed to further improve the

precision and reliability of the results. In particular, the efforts for a future effective operation of the speech recognizer in the reliability-oriented field of maritime search and rescue require that all essential information is interpreted error-free by the system, i.e., a transcription of the spoken word including all speaker-generated errors is generated true to the original. In an intermediate step, it might be useful to add additional information typical for the sea area, such as the names of prominent geographical formations or possible ports, to the speech recognizer. Furthermore, an optimizing solution is needed to take into account accent-related variations in the pronunciation of ship names.

Initial tests of the localization algorithm in the simulation environment delivered plausible values for the determination of the bearing probability *PBRG*. As the distance between measured baring line and ship position becomes smaller, the corresponding bearing probability increases. The practical testing of the technology and the validation of the established relationships in practical application at sea is planned for the second half of the project. Additions to the current direction-finding algorithm are planned to improve the results of the determined bearing probabilities. This includes measures to increase the positional accuracy of a ship under consideration by taking into account the signal strength of a received radio signal or bearing values of multiple radio direction finders installed at different locations. This also includes taking into account other dynamic ship data, such as the rate of turn, when forecasting the new ship position in order to achieve greater accuracy.

## 5. Patents

The system presented is registered as a European patent under EP 3 664 065 (2020).

**Author Contributions:** Conceptualization, T.L. and A.G.; methodology, T.L.; software, A.M. and I.Z.; formal analysis, M.R., O.J. and J.-G.R.; investigation, A.G., T.L., M.R., O.J. and J.-G.R.; data curation, A.G., J.-G.R., and T.L.; writing—original draft preparation, T.L., A.G., J.-G.R., O.J. and M.R.; writing—review and editing, T.L., A.G., J.-G.R., O.J. and M.R.; visualization, A.G., J.-G.R., M.R., O.J. and I.Z.; project administration, T.L.; funding acquisition, T.L. All authors have read and agreed to the published version of the manuscript.

**Funding:** This research was funded by the German Federal Ministry of Education and Research (BMBF) under the Research Program for Civil Security of the German Federal Government, grant number 13N15018, 13N15019, and 13N15020. The APC was funded by the German Federal Ministry of Education and Research, grant number 13N15018.

**Acknowledgments:** We would like to thank Celien Bosma and David Laqua for their support in conducting the workshop. In this context, we also express our special thanks to all participants and end users of the workshops for sharing their experiences and ideas with us. We are especially grateful for our associated partners for accompanying the ARTUS project, and the programming contributions of Emin Nakilicoglu.

**Conflicts of Interest:** The funders had no role in the design of the study; in the collection, analyses, or interpretation of data; in the writing of the manuscript, or in the decision to publish the results.

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
