# Peer review of "Assisting Maritime Search and Rescue (SAR) Personnel with AI-Based Speech Recognition and Smart Direction Finding"

_jmse, doi:10.3390/jmse8100818_

Round 1

Reviewer 1 Report

Valuable work, matched to the subject of the journal.

Remarks:

  1. Please change the position of Figure 2 on page 6 - the caption must be under the figure, on the same page.
  2. It would be worth writing more information about the DF Antenna Unit - the number of antennas and their type and accuracy of determination of position or direction.

Author Response

Dear Reviewer,

Thank you very much for your comments on our manuscript, which have been very helpful in further developing our concept paper. We were happy to implement your comments and suggestions for improvement:

  1. due to the active "track change mode" in Word we unfortunately made this mistake. We have corrected the position of the figure and the caption accordingly.
  2. we gladly followed your suggestion and added further information about the DF antenna unit we use. We hope that this description supports the comprehensibility of our concept (lines 319-336).

Furthermore, in view of the conceptual character of the paper, we have made additions in Section 2 on the further methodological procedure (lines 95-100) and, for the sake of completeness and better comprehensibility, we have referred to our explanations in Section 3.3 (lines 89-94).

Once again, thank you very much for your constructive feedback.

With kind regards on behalf of the authors,
Your Thomas Lübcke

Reviewer 2 Report

The authors addressed all of my comments and I think that this paper is now suitable for publication.

One minor comment: Replace "neuronal" with "neural" on line 244 or in any other text occurrence.

Author Response

Dear Reviewer,

Thank you very much for your valuable input, which has been very helpful in further developing our concept paper. We have added two more additions to the methods section: First, we have added a short outlook on section 3.3. for better readability (lines 89-94). Secondly, we have briefly outlined the work steps still to come in order to give the research design a clearer outline (lines 95-100).

We are very grateful for your advice regarding the correction of the word "neural" - sometimes a German word simply sneaks into the text and as a native speaker you may even read over it several times. We have checked the entire text again. Now everything should be correct.

Thank you again for your constructive feedback.

With kind regards on behalf of the authors,
Thomas Lübcke

This manuscript is a resubmission of an earlier submission. The following is a list of the peer review reports and author responses from that submission.

Round 1

Reviewer 1 Report

The authors present a system which is able to not only locate a vessel in distress but to recognize voice messages sent by those vessels in order to assist officers in SAR operations. Although such a system is a promising idea I failed to see whether the system shows promising results. Regarding the voice recognition part of the system: Figure 2 shows the word-error-rate of three different methodologies and compares the error rate between real and simulated data. The issue here is that there is a lack of description of these methodologies. What is the Adapted Acoustic model, what kind of machine learning methods does it use, if any? The same questions also apply for the Added SMCP and the improved ship names. Furthermore, what methodology was used for training-testing, e.g. 80-20 train-test split? I would like to see a k-fold cross validation to further validate that the authors' model has been appropriately tested. Regarding, the direction-finding algorithm: I would like to see a better description by also adding a pseudocode to this methodology. It is not clear how the algorithm detects the location of the vessel. Which algorithm is used to predict the bearing probability PBRG? Moreover, I did not see any experimental tests showcasing the accuracy of the overall direction-finding algorithm. The authors frequently state that the results are promising but the readers cannot say the same because there is a lack of experimental tests. Finally, the relevant work of the paper is poor. Are there any other assistive applications? What kind of assistance do they provide? Are they similar to the methodology the authors present?

Reviewer 2 Report

The work has the application nature, it describes the realized and patented prototype. This is a big plus of work. However, the scientific load of the work is quite weak. On the one hand, the process of creating requirements and building the system is presented, but there is no information about algorithm of the speech recognition and the algorithm of the position's determinationd. Perhaps these are patent-protected areas. 

Detailed comments:
1. The passive voice must be used (especially line 102 and 105). Chapter 3.1 should be reviewed in this regard.

2. Add number of Figure for lines 343 and 359.

3. Please try to rearrange the text in chapter 3.2. Figure 5 should not be split across pages.

4. The text on lines 270-273 looks quite strange (the translation of the acronym Radar in a scientific journal looks quite strange), but for the proposed journal it may be right.